# Social Media Discourses on Interracial Intimacy: Tracking Racism and Sexism through Chinese Geo-located Social Media Data

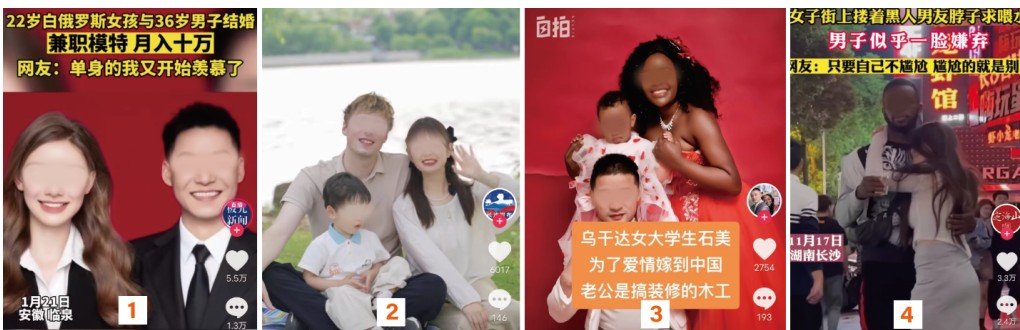

**Figure 1: Four interracial relationship types: (1) White women and Chinese men. (2) White men and Chinese women. (3) Black women and Chinese men. (4) Black men and Chinese women.**

## ABSTRACT

By analyzing the regional differences in the sentiment of comments on short video posts related to interracial intimate relationships on *Douyin*, a Chinese social media platform, we depict the Chinese social media discourses on four interracial relationship types (Black men and Chinese women, Black women and Chinese men, White men and Chinese women, White women and Chinese men) and explore potential regional differences in these discourses. The region information is derived from the IP geolocation, which has been publicly available since April 2022, when the Chinese government mandated social media companies to display the IP geolocation of all platform users. Our content analysis revealed that the Black men and Chinese women couples attracted the most negative comments and the White women and Chinese men couples received the least negative comments. We also observed substantial regional differences in the discourses towards these interracial relationships. We investigated several provincial socioeconomic development indicators and noted that local GDP, population sizes, and the level of openness to Western cultures can explain the variation in the negative sentiment level. This work advances our understanding of the interplay of race, gender, and immigration in constructing public discourses on social media and offers important insights into how these discourses evolve along with socioeconomic development.

## CCS CONCEPTS

• **Human-centered computing** → *Social media*.

## KEYWORDS

Interracial Intimate Relationships, Social Media, Sentiment Analysis, IP Geolocation

**ACM Reference Format:**

Anonymous Author(s). 2018. Social Media Discourses on Interracial Intimacy: Tracking Racism and Sexism through Chinese Geo-located Social Media Data. In *Proceedings of Make sure to enter the correct conference title from your rights confirmation emai (Conference acronym 'XX).* ACM, New York, NY, USA, 10 pages. https://doi.org/XXXXXXX.XXXXXXX

## 1 INTRODUCTION

Interracial intimate relationships are becoming more common in today's increasingly globalized world. In mainland China, international marriages have risen from less than 8,500 in 1979 to over 49,000 in 2010 [20]. Despite the growing number of interracial relationships, in countries with a clearly dominant ethnic group such as China, this type of relationship remains a minority and can be a subject of strong social censorship [3]. The emphasis on ancestry, lineage, and cultural integration in the traditional Chinese culture, intertwined with the political advocacy of Chinese nationalism, makes interracial intimacy less acceptable in China [31, 49, 61].

Discussions about interracial relationships on social media provide critical insights into the discourses on race, gender, and immigration and can significantly influence public opinion and policy [9, 24]. The amount of news and information people consume influence perceptions and behaviors toward immigrants [2]. In this context, *Douyin*, the Chinese *TikTok* offers an ideal setting to explore how these relatively new and rare intimacy types are viewed. In addition, people also tend to be more open to expressing their views and attitudes on the internet [1, 19]. Therefore, analyzing comments related to this sensitive topic on these social media can offer a comprehensive evaluation of the diversity of attitudes in such discussions.

Previous work suffers from the lack of any socioeconomic explanations for these discourses on interracial relationships due to

the lack of any socioeconomic indicator from those who comment online. Our work, in addition to providing more insights about this topic in a Chinese context, advances our current knowledge by linking the user's location with their comments on *Douyin*. In April 2022, the Chinese government mandated social media companies to display the IP geolocation at a provincial level or country level (for overseas users) of all platform users. This universal disclosure of IP geolocation has solved the substantial sample selection problem that platform users proactively choose to disclose their location. Utilizing this feature, we can (1) depict the regional differences of the sentiment based on the IP geolocation of these comments; and (2) link external geo-located socioeconomic development indicators to examine which factors can contribute to explaining the different discourses. Our study aims to fill a critical research gap by examining how sentiment towards interracial relationships on *Douyin* differ in different provinces in China where the socioeconomic development level varies, thereby contributing to our understanding of how race, gender, and immigration intersect in online discourses in China.

In this work, we analyzed *Douyin* users' comments towards short video posts related to four interracial relationship types in China: Black men and Chinese women, Black women and Chinese men, White men and Chinese women, and White women and Chinese men. Our research focuses on three directions: (1) The topics of concern in the discussion of interracial relationships in comments from different IP geolocations; (2) the negative sentiment level toward the different interracial relationship types in comments from different provinces; and (3) the socioeconomic factors that can explain the sentiment differences in these discussions about interracial relationships. We analyzed data from 0.5 million related comments of 549 short videos about interracial relationships to answer the following questions:

- **RQ1: What themes emerge in the comments about interracial relationships?**
- **RQ2: Are there differences in sentiment across the four interracial intimate relationship types?**
- **RQ3: Whether there are any regional differences in the themes and sentiment towards these interracial relationships?**
- **RQ4: What factors can explain the differences in the negative sentiment levels in each of the four interracial relationships?**

We find that posts about Black men and Chinese women attracted the most negative comments, while posts about White women and Chinese men received the least negative comments. Comments are centered around topics about lineage, material gains, and aspirations for a better life. There are also regional variations in the levels of negative sentiment towards these relationships. After investigating several provincial socioeconomic development indicators, we find that local GDP, population sizes, and the level of openness to Western culture explain some of these variations. This study deepens our understanding of how social media discourses on interracial intimacy are shaped by racism and sexism. The findings also provide more nuanced insights into how these discourses evolve with socioeconomic development that differs in the various regions in China.

## 2 RELATED WORK

We first offer some background information about interracial intimate relationships in China. We then introduce the interracial intimacy triangle theoretical model. Finally, we mention briefly the policy regarding IP geolocation disclosure on Chinese social media platforms to assist the understanding of our later analysis.

### 2.1 Immigrants and Interracial Relationships in China

In China, interracial relationships were very rare before the early 1990s [20]. Although the Chinese population is composed of 56 ethnic groups, the Han Chinese account for 91.59% of the overall Chinese population [57]. In addition, the physical characteristics of these ethnic groups in China do not differ much from each other. Therefore, the concept of "race" was alien to most Chinese, and interracial relationships were of limited attention until China completely opened its border to the rest of the world. Even in the early 21st century, despite the increased number of international marriages, the proportion of this group of all marriages was still low, with the more developed eastern coastal cities witnessing most of them [20].

In addition, the stringent immigration policy makes China a not ideal immigration destination [16]. Migrants incur a high cost to perform transnational long-distance migration to China with the hope to obtain higher returns on capital, and improve their socioeconomic status [12]. Many of them stay even when their Chinese visa expires. In the US, cultural stereotypes lead to negative perceptions of undocumented immigrants among white Americans [10]. Stereotypes about immigrants often encompass a range of fears, including the belief that they may contribute to crime rates, create self-segregated communities, exacerbate economic difficulties, and demonstrate disloyalty to the host country [11, 50]. Therefore, in China, international marriage is often viewed with suspicion and concern [40].

Some common concerns are the serious cultural shocks or conflicts that can result in issues within a marriage or marital dissolution [48]. Some might downplay the significance of this problem with reference to the U.S., where interracial intimacy is more common. However, the situation there is quite different since the growth in intermarriage has coincided with shifting societal norms and the self-labeling of the U.S. to be an immigrant country. Americans have become more accepting of marriages involving spouses of different races [33].

The "gender tension" in the marriage market further complicates the seemingly negative sentiments towards interracial relationships. The son preference of the Chinese family culture has led to a serious gender imbalance with a skewed male-to-female ratio. China is still a society with universal marriage, and people have a strong desire and pressure to get married at a proper age. Men who stayed single for a lifetime were stigmatized and ridiculed. Therefore, on the one hand, some Chinese men may view a large quantity of male immigrants who come to China are competing with them in the marriage market, making the chance to get married even lower [61]. On the other hand, cross-border marriages with brides from overseas can be a possible solution to deal with the male-heavy marriage market [58].

Overall, interracial intimacy is uncommon in China, concentrates in certain coastal regions in China, and is a highly controversial topic considering its societal implications.

## 2.2 Theoretical Models

We propose a triangular theoretical framework to explain people's perceptions of interracial relationships. This framework combines patriarchal racism, marital exchange, and immigrant integration theories to explain how Chinese platform users interpret the racial and gender characteristics of intimate relationships.

*2.2.1 Patriarchal Racism.* The construction of the concept of "race" is influenced by the historical binary opposition of racial inferiority/superiority [3, 53]. Modern theorists of racism constructed a "Racial hierarchy" that is composed of eugenics and racial development [13]. It places White and Asian people at the top, where the Chinese civilization is presented as culturally superior [18], and other races are regarded as inferior [8]. Similar to in other countries, in China, racism is represented by the strong anti-black ideology where black people are often viewed to be inferior to Chinese and Whites.

Views about migrants are often based on gender stereotypes [26]. Therefore, racism is embedded in the patriarchal system and is referred to as patriarchal racism [36]. Local communities tend to hold gender biases against migrant groups. For example, young male immigrants are portrayed to pose a security and cultural threat to local communities [55]. Young, male immigrants are less welcome in the host country than women immigrants. [30, 46]

In addition, as briefly mentioned above, the sex imbalance in the marriage market shapes people's views about interracial intimacy. Female immigrants, especially from low-income countries, may help alleviate the squeezed marriage market for Chinese men, and male immigrants work as competitors in the marriage market for Chinese men. It is clear that the perceived threat to the orthodoxy of traditional culture significantly influences Chinese public opinions on immigration [61]. Chinese social media users may worry that immigration and interracial marriage will wipe out Chinese lineage and culture [32].

The patriarchal racism is likely to lead to a skewed view that favors female immigrants. This bias is further complicated by the anxiety of losing the "culture/lineage purity" when Chinese women marry non-Chinese individuals, reinforcing the patriarchal and racial prejudices [61]. Particularly in the case of black immigrants, who are often subjected to racial bias and discrimination. The public's concern about maintaining cultural purity and safety, fueled by the anti-black notion and the skewed sex ratio in the marriage market, has led to a decreased acceptance of black male immigrants and resistance to liberal immigration policies.

*2.2.2 Marital Exchange.* The "marriage exchange theory" was born from the phenomenon of interracial marriages between Blacks and Whites in the United States, where relationships between Blacks with higher socioeconomic status and Whites with lower socioeconomic status are more common [44]. This phenomenon is described as an exchange regarding race and economic gains [37] as marriage maximizes its function through specialization and exchange [60].

This exchange model also applies to interracial relationships in China. For example, Chinese people, especially females, may marry people of a different race in exchange for material gains or residency status in another country. African businessmen in Guangzhou often rely on close social relationships with Chinese women [29]. These women play a key role in acquiring resources and building trust with local business partners, thereby reducing transaction costs and laying the foundation for business success [23]. Therefore, the marriage exchange theory can be used to explain the phenomenon of groups with different characteristics such as appearance, ethnic identity, and socioeconomic characteristics establishing intimate relationships. People may therefore think that those individuals, especially Chinese women, who marry Blacks or Whites are mainly for monetary gain or eugenic reasons. This theory supports the hypothesis that people may view interracial relationships between Chinese women and male immigrants more negatively compared to those between Chinese men and female immigrants.

*2.2.3 Immigrant Integration.* In the past few decades, with China's rapid economic growth, investors, international students, global professionals, and businessmen from different regions around the world have been attracted to China [17]. Attracting foreign investment for local development has been the central goal of local government. Guangzhou City, located in the Guangdong Province of China, for example, has now become the home to the largest community of African immigrants in Asia [54]. Based on the immigrant integration theory, when facing a strong mainstream culture, migrants will choose different adaptation strategies to achieve cultural integration and experience the process of "contact-competition-assimilation-coexistence-integration" [56]. Marriage is one of the most effective pathways to achieving the cultural integration of immigrants. Therefore, Chinese people may think that those who marry a spouse of a different race can work as Chinese culture ambassadors and advocate Chinese culture.

## 2.3 Social Media Analysis with IP Geolocation

Many studies have utilized the geolocation information of platform users to link local information with user behaviors on social media. For example, researchers analyzed public responses to political campaigns [43], online advertising [41], and vaccinations [34]. The biggest issue related to these studies is that the users can decide whether to disclose their geolocation or not, and they disclose for various purposes. For example, users use geographical positioning on social media to find new contacts [38, 51] and disclose their geolocation to showcase interesting places and display a personal image of popularity and engaging [15, 22]. Some users choose to geo-tag when sharing products on social media to receive discounts or gifts [25, 59]. Many users do not disclose for privacy and security considerations [28, 45]. Therefore, those who disclose are likely to differ from those who do not. Previous research has found that in posts with disclosed IP location, the sentiment is happier than those not disclosed [22]. Therefore, researchers can only analyze a highly selective sample, and we are not sure whether any findings derived from this group can be generalized to those who do not disclose their geolocations.

Starting in April 2022, the Chinese government requires social media platform owners to make the provincial IP geolocation of

their users public. Therefore, all comments on the Chinese social media platforms are geo-tagged at the provincial level (by country or overseas users). This move is intended to make internet users' information more transparent and the build a more-regulated network. This universal disclosure policy enhances the generalizability of the relationship between local socioeconomic development and comment sentiment toward interracial relationships in our study.

## 3 DATA

The platform users' comment data is from the Chinese social media platform *Douyin*, one of the most popular short video social media with more than 1 billion registered users in Mainland China and over 700 million active daily users [42]. The data crawling period is from September 3rd to September 23rd, 2023, with 14 hours of data crawling per day. The cleaned data consists of 549 short videos about interracial relationships and 501,683 related comments with user IP geolocation from May 2022 to September 2023 across all 31 provincial administrative regions in Mainland China. The raw data and Stata code for statistical analysis are on Github[1].

### 3.1 Video posts and comments extraction and clean

We are interested in four interracial relationship types: White men and Chinese women, White women and Chinese men, Black women and Chinese men, and Black men and Chinese women. We set 12 keywords in each type, such as "White husband" or "Black honey" for search and extraction, as shown in Appendix A.1. After we had extracted 895 short videos with keyword search, three researchers manually checked these videos to ensure their conformity with the theme of interracial intimacy. We excluded 131 short videos that did belong to any of the four types of interracial relationships or were duplicates. We then performed deep artificial annotation to classify short videos into four types.

For extracting comments related to these short video posts, we developed a web crawler based on the open-source tool MediaCrawler[2]. We removed characters indicating username (@+username) in the comment, but we kept the emoji category in the comments because the emoji can indicate the content sentiment.

### 3.2 Simulating crawls from multiple locations

To avoid variations in recommended content caused by the recommendation algorithm based on users' IP geolocation, we used IP proxies provided by chinapptp[3] to simulate IP geolocation from 21 different provinces in China, covering 81.47% of its population. We randomly selected a city within each of the 21 provinces as a representative. The raw data contains 3.57 million comments. We did not use resident IPs to avoid disrupting any individual's online activities.

[1] https://github.com/IRRsocialMedia/IRR
[2] https://github.com/NanmiCoder/MediaCrawler
[3] http://www.chinapptp.com/cmp/webservice?bs=7

## 4 METHOD

### 4.1 Topic Analysis

RQ1 is about the themes that emerged in the comments. We used the BERTopic [14] to categorize text with the same theme label. Previous studies have also used topic modeling to analyze social media data [21]. With our dataset, BERTopic grouped the 0.5 billion comments into 100 topic clusters. We then scoped our investigation to specific clusters that are directly related to interracial relationships. For this purpose, three researchers manually verified the common keywords contained in each cluster. We removed 26 clusters unrelated to the topic, leaving 74 clusters, corresponding to 549 short videos and 501,683 comments. This forms the final analytical sample.

We conducted a qualitative analysis of the 74 clusters and further scaled them down to 16 clusters with synonymous words grouped together. The 16 clusters were manually labeled according to the three theoretical perspectives: Patriarchal Racism, Marital Exchange, and Immigrant Integration. We analyzed 16 clusters to depict the main themes, divided by the geolocation of the comments. The 16 clusters and their grouping into the three theoretical perspectives are reported in Figure 2.

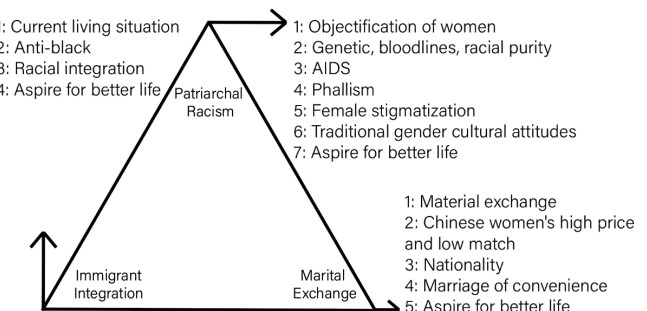

**Figure 2: Correspondence of 16 clusters to three theories**

### 4.2 Sentiment Analysis

RQ2 is about the sentiments reflected in the comments. We fine-tune the roberta-base-cold model for this sentiment analysis. We randomly selected 3000 comments for human annotation and used this labeled dataset to construct a manual annotation instruction. Roberta-base-cold is a sentiment classification model based on the chinese-roberta-wwm-ext [5, 6] sentiment classification model, which utilizes the Chinese Offensive Language Detection Dataset [7] for fine-tuning. The Chinese-roberta-wwm-ext is a baseline model trained on Chinese corpora using RoBERTa-base. The fine-tuned model categorizes comments into Hate and NO Hate categories and demonstrates excellent performance on our test set.

Table1 presents the result of the sentiment analysis across the four interracial intimacy types. First, most video posts are about the relationship between Chinese women and Black/White men. Videos about White women and Chinese men are the most uncommon. The highest level of sentiment with the largest proportion of comments being classified as "Hate" is related to the Black men and Chinese women match.

**Table 1: Sentiment analysis summary. BM & CW - Black men and Chinese women, Bw & CM - Black women and Chinese men, WM & CW - White men and Chinese women, and WW & CM - White women and Chinese men.**

|  | BM &CW | BW &CM | WM &CW | WW &CM | Total |
|---|---|---|---|---|---|
| Number of Video Posts | 192 | 134 | 186 | 37 | 549 |
| Number of Comments | 185779 | 146164 | 145512 | 24228 | 501683 |
| Sentiment: No Hate | 62486 33.6% | 68609 46.9% | 69758 47.9% | 13686 56.5% | 214539 42.8% |
| Sentiment: Hate | 123293 66.4% | 77555 53.1% | 75754 52.1% | 10542 43.5% | 287144 57.2% |

## 4.3 Regional Differences

RQ3 asks about whether these sentiment analysis results differ for comments from different provinces. We, therefore, conduct all the above analyses by the IP geolocations of the comments.

## 4.4 Socioeconomic Development Explanations

RQ4 is about what factors may explain the differences in sentiment in different regions. We employed linear probability regressions [4] to explore the relationship between the sentiment in comments and the geolocation-linked variables. We control for the short video IDs, thereby eliminating the issue that people located in different areas are viewing different video posts due to algorithms. The model shows whether, for the same short video posts, comments from different geolocations (with different socioeconomic development indicators) demonstrate different sentiments.

The matched provincial-level socioeconomic indicators include the GDP per capita, the Human Development Index (HDI), which is a composite indicator based on life expectancy, years of schooling, and Gross National Income per capita [52], reflects regional socioeconomic development levels. The sex ratio, calculated by the population size of males over females in the province, reflects the level of patriarchy. Population size is also selected to reflect the level of competition in the local labor market. We also employ the number of Starbucks stores and the number of foreign people to reflect cultural openness to Western culture and the international immigration level, respectively. All the data is from China's Seventh National Population Census. The regression analysis aims to identify which provincial-level factors can explain the differences in the negative sentiment levels.

## 5 RESULT

## 5.1 Main Themes and Regional Differences

Figure 3-(1) displays the word cloud of 16 words emerging from the topic clusters. The figure is generated based on the words' higher weighted log odds [39]. Figure 3-(2) (3) (4) and Figure 2 shows how the 16 topic clusters are grouped into three theoretical perspectives – patriarchal racism, marital exchange, and immigrant integration. The Chinese version is in Figure 8 in Appendix A.3.

*5.1.1 Aspire for a better life.* One topic cluster, "aspire for a better life", appears in all three theoretical frameworks. This is a term that is considered to be a positive sentiment of admiration and blessing for the Chinese partners in the comments. In the group "Aspire for a better life", the most common words in the comments included "Roses", "Congratulations", "Thanks", "Beautiful", "Cute" and "Warm".

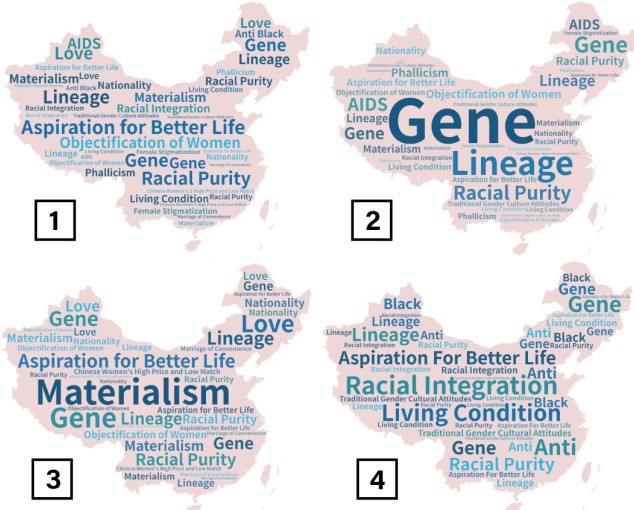

**Figure 3: Word cloud map of (1) all 16 words; (2) 7 words in patriarchal racism theory; (3) 5 words in marital exchange theory; (4) 4 words in immigrant integration theory.**

*5.1.2 Patriarchal racism: Anti-black sentiment.* Figure 3-(2) reports seven clusters in this perspective, namely 1: "Objectification of women", 2: "Genetic, bloodlines, racial purity", 3: "AIDS", 4: "Phallism", 5: "Female stigmatization", 6: "Traditional gender cultural attitudes" and 7: "Aspire for a better life". The "Genes, lineage, racial purity" group was the most common, mostly lined to posts about Black men marrying Chinese women. In this topic group, comments usually included "Don't go back to our country", "Genes", "Lineage", "Blacks", and "Tears", reflecting a strong anti-black sentiment. Many of these comments are specifically targeting Chinese women. They reflect the objectification of women as nothing but mating partners. The majority of the comments were from Guangdong Province, with 45,968 comments, followed by Jiangsu Province, Henan Province, Shandong Province, and Zhejiang Province (Figure 4). This is proportional to the distribution of the Chinese population by province.

*5.1.3 Marital exchange: Material gains.* The 5 words in this perspective in Figure 3-(3) are 1: "Material exchange", 2: "Chinese women's high price and low match", 3: "Nationality", 4: "Marriage of convenience" and 5: "Aspire for a better life". The "Material exchange" cluster is the most common, and all four types of interracial relationships are linked to these words similarly. In the "Material exchange" group, comments most often included phrases like "Each takes what he needs", "Own a house and a car", "Sober in the world", "Bride price", and "Applauding", reflecting the importance and attention given by the comments to the material gains of marriage.

The largest number of comments was in Guangdong Province, with 34,125 comments (Figure 4).

*5.1.4 Immigrant Integration: Chinese culture.* Figure 3-(4) reports the 4 topic words belonging to this perspective, namely 1: "Current living situation", 2: "Anti-black", 3: "Racial integration" and 4: "Aspire for a better life". In the "Racial Integration" group, comments most often included words such as "Like", "Love", "Chinese language", "Neighbor", and "Accent", reflecting positive attitudes and recognition of racial integration and cultural exchange. However, in the "Anti-black" group, there were comments expressing distaste for black immigrants.

**Table 2: Distribution of the three theoretical perspectives across interracial couple types**

| Couple Types | Patriarchal Racism | Marital Exchange | Immigrant Integration |
|---|---|---|---|
| 1:BM&CW | 63.6% | 47.3% | 19.9% |
| 2:BW&CM | 70.7% | 50.1% | 17.4% |
| 3:WM&CM | 63.5% | 52% | 18.9% |
| 4:WW&CM | 68.1% | 49.4% | 17.2% |

Table 2 reports the distribution of the three theoretical perspectives across the four interracial relationship types. We label the comment based on the above discussions about topic clusters and theoretical labeling. One comment may belong to multiple theoretical frameworks. In all four types, patriarchal racism plays the central role, especially when the relationships involve Chinese men. The marital exchange perspective is more important when the relationships involve Chinese women. Immigration integration is also more common when the relationship is between immigrants and Chinese women.

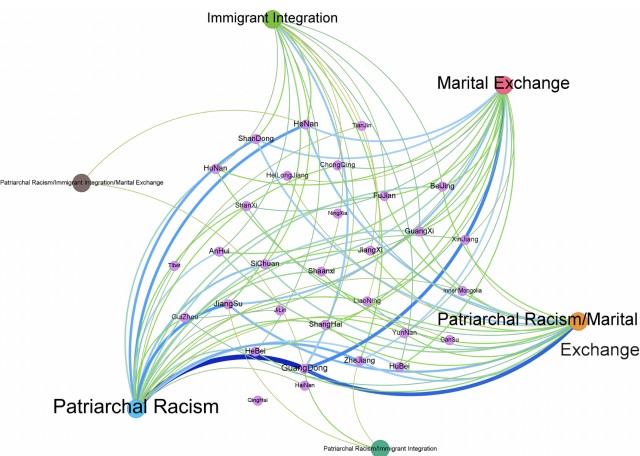

**Figure 4: Network of IP Geolocation and Theoretical Models**

Figure 4 displays the link (the edge) between the themes (one comment can belong to multiple themes) and the geolocation of the comments. The thickness of these edges corresponds to their weights. A thicker edge implies more comments belong to this link. For clarity and emphasis on key relationships, only nodes and edges with at least 1,291 comments were retained, preserving 52.15% of the total comments. This figure summarizes all these discussions above with better visualization of regional differences in topic themes. For example, Guangdong, being the home to the largest community of African immigrants in Asia, demonstrates a large quantity of comments linked to patriarchal racism and martial exchange, with limited reference to immigrant integration.

## 5.2 Sentiment and Regional Differences

We found significant differences in negative attitudes towards different interracial relationships and in comments from different provinces as shown in Figures 5 (1)-(4).

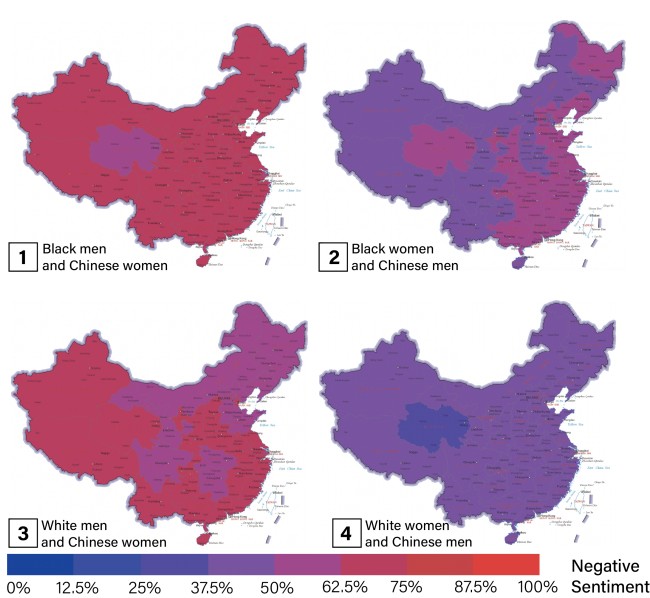

**Figure 5: Level of negative sentiment by province**

*5.2.1 Black men and Chinese women.* Figure 5 (1) shows the highest level (66.4% ) of negative sentiment in those comments towards the Black men and Chinese women couples among the four interracial couple types. The regional variations in this sentiment are small. Guizhou province showed the most negative sentiment towards this type of interracial relationship, with 69.22% classified as negative sentiment, followed by Hunan and Henan provinces. Comments from Qinghai Province and Shanghai City hold the least negative attitude, but the percentage of negative sentiment is still as high as 60%. Qinghai is the province with the highest percentage (46%) of ethnic minorities, while Shanghai is one of the most developed areas in China.

*5.2.2 Black women and Chinese men.* Figure 5 (2) shows a much more neutral sentiment than that towards the first type, with an average of 53.10% of negative sentiment towards such interracial relationships. Guangxi province showed the most negative (57.17%) sentiment, and the Tibet Autonomous Region is the provincial administrative region with the least negative (43.11%) attitude.

*5.2.3 White men and Chinese women.* In Figure 5 (3), 52.10% comments are classified to present negative sentiment towards such interracial relationships on average. Guizhou province and Yunnan showed the most negative sentiment with over 55% classified to be negative sentiment. Heilongjiang province has the lowest negative (47.74%) sentiment.

*5.2.4 White women and Chinese men.* Figure 5 (4) shows the least negative sentiment (43.5%) compared to those towards the rest of interracial relationship types. Also, there is little difference in the sentiment across regions, except for Qinghai Province, which has the lowest negative attitude level (35.29%). Again, being the province with the highest percentage (46%) of ethnic minorities, people in Shanghai may be more familiar with inter-racial and inter-ethnic marriages.

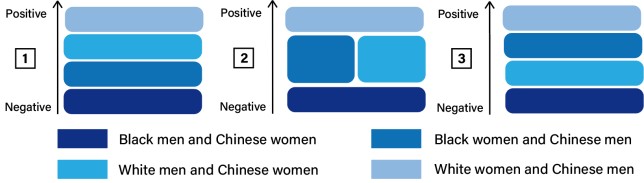

**Figure 6: The three types of sentiment ranking: (1) Type 1 - More Racist. (2) Type 2 - More Exchange. (3) Type 3 - More Patriarchal.**

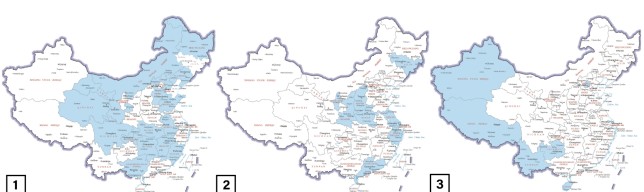

**Figure 7: Provinces belong to the three sentiment ranking types**

We further categorized the 31 Chinese provinces into 3 types based on the ranking of the four international relationships by sentiment level and report them in Figure 7. Please see Table 5 of Appendix A.2 for more details.

**Type1: More Racist** is characterized by the more negative sentiment toward interracial relationships that involve Blacks and less negative sentiment toward interracial relationships that involve Whites with male immigrants ranked lower than female ones. 16 provinces belong to this group, such as Shandong, Heilongjiang, and Sichuan, as shown in Figure 7 (1).

**Type2: Exchange** features those who have similar sentiments towards Black women and Chinese men couples and White men and Chinese women couples. This may imply users' views that those Chinese men are marrying down for gains and those Chinese women are marrying up to gain a better life. This type includes Shanghai, Henan, Guangdong, and nine other regions, as shown in Figure 7 (2).

**Type3: More Patriarchal** is characterized by a more negative sentiment attitude towards interracial relationships involving Chinese women married to a different race, and a more positive sentiment attitude towards those involving Chinese men. Six provincial administrative regions, including Yunnan Province and Beijing, are characterized by this type. Apart from Beijing, these areas are mainly concentrated in Western and Southern China with disadvantaged socioeconomic development (Figure 7 (3)).

## 5.3 Socioeconomic Development Explanations

Table 3 reports the result from a multi-variant linear probability model (fixed on the short video post IDs) to predict the probability of having a negative comment towards each of the four interracial relationship types.

**Table 3: Predicting negative sentiment likelihood, by interracial relationship types**

| Type | BM&CW | BW&CM | WM&CW | WW&CM |
|---|---|---|---|---|
| Per capita GDP/1000 | 0.0478*** (0.00885) | -0.00320 (0.0108) | 0.0215** (0.0107) | 0.00454 (0.0270) |
| Male/Female ratio/10 | -0.00292 (0.00349) | 0.0282*** (0.00419) | 0.00139 (0.00420) | -0.00865 (0.0106) |
| Human Development Index | -0.0770 (0.0546) | -0.141** (0.0666) | 0.0400 (0.0643) | 0.158 (0.159) |
| Log(Population -size)/10 | 0.130*** (0.0238) | 0.0205 (0.0286) | 0.0712** (0.0284) | 0.0455 (0.0707) |
| Log(Foreign -people)/10 | -0.0216* (0.0124) | -0.00462 (0.0159) | 0.0181 (0.0153) | 0.104*** (0.0384) |
| Starbucks/100 | -0.208*** (0.0404) | 0.165*** (0.0500) | -0.112** (0.0495) | -0.298** (0.127) |
| Constant | 0.514*** (0.0671) | 0.301*** (0.0829) | 0.320*** (0.0801) | 0.240 (0.203) |
| Short video ID-fixed-effect | Yes | Yes | Yes | Yes |
| Observations | 185,779 | 146,164 | 145,512 | 24,228 |
| R-squared | 0.086 | 0.103 | 0.092 | 0.033 |

*5.3.1 Black men and Chinese women.* Under the same short video, comments from areas with a higher GDP per capita and more populations display more negative sentiments towards interracial relationships between black men and Chinese women. Meanwhile, areas with more foreign people and more Starbucks stores show less negative attitudes towards this group. Therefore, *Douyin* comment makers in economically more developed areas with more population may worry that Black people are reaping economic gains from China. Black males are viewed as competitors with Chinese in the local labor market. Comments from areas that are more open to transnational immigration or Western culture show less racist sentiment.

*5.3.2 Black women and Chinese men.* Under the same short video, comments from areas with a higher provincial sex ratio (men to women) show a higher chance of endorsing negative sentiment. On the other hand, comments from areas with a better human

development level are less negative towards this group. The number of Starbucks stores is positively linked to negative sentiment probability.

*5.3.3 White men and Chinese women.* Again, Under the same short video, comments from areas with better economic development and a bigger population display more negative attitudes towards this group. Areas with more Starbucks stores show fewer negative comments.

*5.3.4 White women and Chinese men.* Regarding the last interracial relationship type, the size of the foreign population is statistically significantly linked to the probability of having a negative comment. Comments from areas with more foreigners tend to display a more negative sentiment. The number of Starbucks stores still has a negative correlation with the negative sentiment.

## 5.4 Discussion

With China's fast economic development and the strong demand for foreign investments, more and more international immigrants come to China, and some of them form intimate relationships with Chinese people [47]. Meanwhile, a soaring number of foreigners also come to China to study, work, and live, leading to their increasing interactions with Chinese people [35]. Nonetheless, this group of interracial relationships is still rare and is viewed to be unconventional and undesirable. This paper analyzes comments about the short videos posted on the social media platform *Douyin* to identify differences in attitudes towards four types of interracial intimate relationships. In particular, we utilize the publicly available and universally disclosed IP geolocation to explore potential socioeconomic development indicators that may explain variation in these attitudes on social media.

## 5.5 Suggestion

Social media companies play an important role in today's society. They provide not only a platform for information exchange but also a vital tool to influence and shape public perceptions. The Chinese social media platforms *Xiaohongshu*, *Douyin*, and *Weibo* all have published community standards and guidelines banning content promoting racial or ethnic hatred and discrimination. However, such content is still ubiquitous on these social media platforms. Therefore, for sensitive topics such as interracial intimacy, social media companies have the responsibility and should strengthen the management and guidance of such relevant content. Our research reveals significant public sentiment differences towards interracial intimacy, potentially escalating societal racial divisions. Hence, social media platforms need stringent content management to prevent racial discrimination and maintain fairness. Additionally, the varying acceptance levels of interracial relationships among social media users suggest a need for improved public understanding and acceptance of such relationships.

## 5.6 Limitation

There are several potential limitations caused by data bias. As the IPs of Chinese social media users are only displayed at the provincial level, some people may worry that the conclusions we draw will be limited by IP geo-tag precision. But we believe this will only have limited influence on the results. Our sample comes from China's largest short video social media, with more than 700 million daily active users. In addition, our method of collecting comments is based on *Douyin's* recommendation mechanism. In order to avoid algorithmic differences caused by IPs under the mechanism, we simulate the IPs of varied provinces to ensure that we receive more integrated data. Some comments may be spam discussions that are irrelevant to the content, but we believe that the sample size of our data set is large enough to offset the interference these comments might cause.

Another possible bias is that we only collected *Douyin* short videos based on the identification of 48 keywords, and these short videos literally involve the theme of interracial intimacy. The data we collected may not cover all potentially related but not specifically stated ones. Moreover, the interracial intimacy in our study is only derived from the content displayed on the *Douyin* platform, which cannot represent the sentiment differences of the entire group towards interracial intimacy. In addition, based on previous research on the differences between male and female user comments [27], we surmise that most of the comments we analyzed were likely left by male platform users. Moreover, as stated in Inara et al.'s study [19], the comments associated with our analysis may have been intentionally written by some people in order to attract attention.

Therefore, we would like to underscore that the findings do not represent the attitudes of the Chinese population or even the *Douyin* platform users. The findings only apply to the publicly available comments on *Douyin*.

## 6 CONCLUSION

The lack of quantitative research on the variance in sentiments of Chinese social media users towards interracial relationships drives us to investigate the discourses related to these relationships on social media., Overall, we found that the comments mainly focused on the topics of "Aspire for better life", "genes", "genetic, bloodlines, racial purity", "material exchange", and "cultural integration". Findings from the sentiment analysis lend strong support to the prevalence of anti-black and sexist content on social media related to these topics. Socioeconomic development, especially local GDP and openness to international migrants and Western cultures, plays different roles in shaping the sentiment of these comments in different types of interracial relations. Overall, comments from areas with more developed economies, more population, and less openness to Western cultures tend to display more negative sentiment. We call for further studies to investigate the underlying mechanisms and more effective online regulations on hate speech.

### ETHICAL STATEMENT

We prioritize user privacy and safety in our research, particularly when handling sensitive geolocation data. We do not infer user mobility or misuse data but only associate it with comments. Our study aims to understand public discourse on interracial relationships, not to stereotype or stigmatize specific groups of individuals. We ensure that all data used is anonymized for user confidentiality. This project has been approved by our IRB, guaranteeing our adherence to ethical standards.

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

# A    APPENDIX

## A.1    48 Keywords for Search and Extraction

Table 4: Keyword Search and Extraction for Each Interracial Relationship Type

| Type | White men Chinese women | White women Chinese men |
|---|---|---|
| **Keyword** | White hubby | White wife |
| | White boyfriend | White girlfriend |
| | White husband | White honey |
| | Western husband | Western wife |
| | Western hubby | Western honey |
| | Western boyfriend | Western girlfriend |
| | Date a white man | Date a white woman |
| | Date a Westerner | Date a Westerner |
| | I love white people | I love white people |
| | I love Westerners | I love Westerners |
| | Marry a white man | Marry a white woman |
| | Marry a Westerner | Marry a Westerner |
| | **Black women Chinese men** | **Black men Chinese women** |
| | Black wife | Black hubby |
| | Black girlfriend | Black boyfriend |
| | Black honey | Black honey |
| | African wife | African husband |
| | African honey | African hubby |
| | African girlfriend | African boyfriend |
| | Date a Black woman | Date a Black man |
| | Date an African | Date an African |
| | I love black people | I love black people |
| | I love Africans | I love Africans |
| | Marry a Black woman | Marry a Black man |
| | Marry an African | Marry an African |

## A.2    Three Types of Distribution in 31 Provincial Administrations

Table 5: Interacial Relationships Classification of Sentiment Types

| Classification of Sentiment Types | Includes Provincial Administrations |
|---|---|
| Type1 | Inner Mongolia Sichuan Anhui Shandong Guangxi Jiangxi Qinghai Hubei Hebei Fujian Hunan Liaoning Heilongjiang Gansu Chongqing Tianjin |
| Type2 | Shanghai Jilin Ningxia Shanxi Guangdong Jiangsu Henan Zhejiang Shaanxi |
| Type3 | Beijing Yunnan Xinjiang Hainan Tibet Guizhou |

## A.3    Overview of Word Clouds

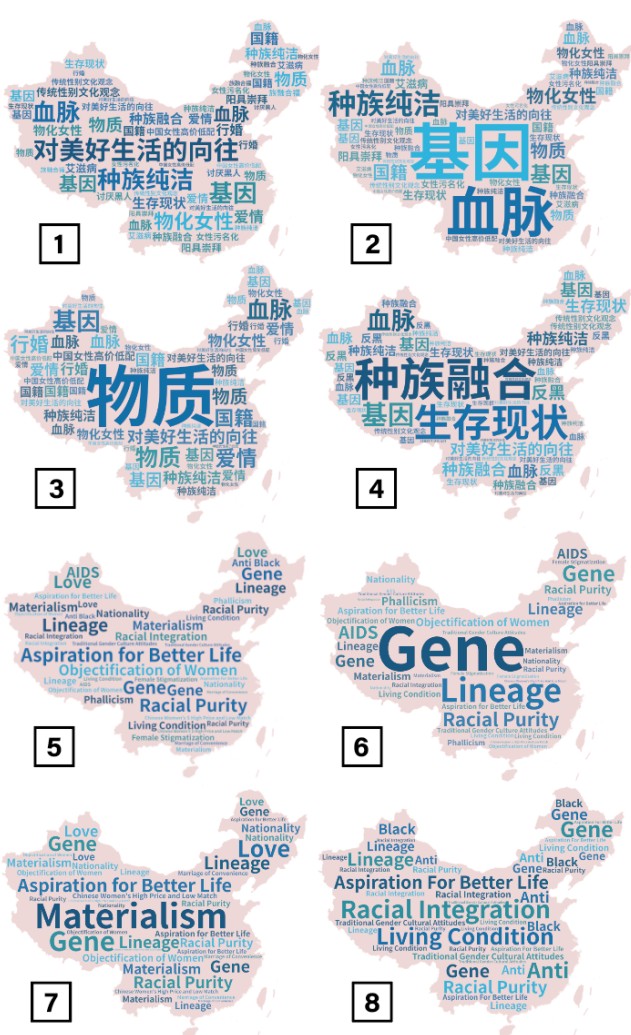

Figure 8: The 16 words associated with discussions on interacial relationships: (1) Word cloud map of all 16 Chinese words. (2) Word cloud for 7 Chinese words in the dimension Chinese patriarchal racism. (3) Word cloud for 5 Chinese words in the dimension marital exchange. (4) Word cloud for 4 Chinese words in the dimension immigration theory. (5) Word cloud map of all 16 words. (6) Word cloud for 7 words in the dimension Chinese patriarchal racism. (7) Word cloud for 5 words in the dimension marital exchange. (8) Word cloud for 4 words in the dimension immigration theory.

