# OpenReview forum: "Social Media Discourses on Interracial Intimacy: Tracking Racism and Sexism through Chinese Geo-located Social Media Data"
_ACM.org/TheWebConf/2024/Conference — TheWebConf24 Oral_

### Official Review · Reviewer_Tvy4 · 2023-11-22

**Novelty:** 6
**Technical Quality:** 6

**Review:**

This study explores the correlations between geographic socioeconomic factors and sentiment toward interracial relationships through analysis of posts on Chinese social media platforms. The study is original and explores an important area in Chinese social media landscape, made possible through the mandate in April 2022 of IP geolocation data being required for all social media posts. It offers a novel contribution in sentiment analysis paired with ip address related geolocation data.

**Questions:**

How reliable is the geolocation data? In other contexts this data has been shown to be rather unreliable. The bulk of this study assumes that the geolocation data is accurate, but this should be substantiated more. What about IP spoofing? VPN use? Bots? Even the reviewers themselves use a service in order to make their IP address appear to be in the particular regions they are studying (line 394). Etc. Consider citing any available resources that demonstrate the reliability of this geolocation / IP information.

**Reviewer Confidence:**

3: The reviewer is confident but not certain that the evaluation is correct

**Scope:**

4: The work is relevant to the Web and to the track, and is of broad interest to the community

---

### Official Review · Reviewer_kRMS · 2023-11-24

**Novelty:** 4
**Technical Quality:** 5

**Review:**

As a non-subject-matter expert, the evaluation of this study is being limited only to the technical aspects of the paper.


Topic modeling:
- How are the hyperparameters chosen? The results of running BerTopic could be sensitive to the choices of hyperparameters. How can the authors justify their choice on the hyperparameters?

- Also, in a similar vein, the results of Bertopic are often sensitive to random initialization. Have the authors attempted to run Bertopic with different random seeds?

- It would be informative to provide top-k words that appeared in each topic so that the reviewers/readers can see how the authors came up with the topic names.

Sentiment analysis:
- Similar comments from the topic modeling can be also applied here. Could the authors provide information on how sensitive the model is on hyperparameters and random seeds?
- Regarding the annotation process: could the authors provide more information on the annotation process? How the annotators are trained to perform the task?
- Has the model been validated on the held-out validation set? How do you know if the model produces accurate results or not?

As Section 5.1 and 5.2 are based on the topic modeling and the sentiment analysis, it is hard to assess the value of those sections.

5.3. What is multi-variant linear probability model? And can the authors elaborate on how to interpret the table (Table 3)?

Although the paper studies a topic that could be considered important and has an academic value, the current version of the paper does not seem to have an academic rigor in executing the experiment, relying on rather anecdotal evidence.

**Questions:**

Please see the questions above.

**Ethics Review Description:**

It is not certain if this study requires ethics review, but would like to let the PCs know that the study involves some information that can potentially be private.

**Reviewer Confidence:**

3: The reviewer is confident but not certain that the evaluation is correct

**Scope:**

4: The work is relevant to the Web and to the track, and is of broad interest to the community

---

### Official Review · Reviewer_fAja · 2023-11-24

**Novelty:** 5
**Technical Quality:** 2

**Review:**

The paper examines discussions of four interracial relationship types (BM-CW, BW-CM, WM-CW, WW-CM) focused on Douyin. It capitalizes on access to IP geolocation data to shed light on regional differences in users' attitudes and relevant their correlates such as GDP, population, and openness to Western cultures at an aggregate level.

Pros
* Immigration and xenophobia in Asia, relative to Western countries, is understudied
* Use of geolocations data (owing to the government mandates) alleviates the problems related to sample selection

Cons
* The process through which the authors narrow the topic categories to the 16 big clusters and matching them with the three-fold theoretical framework is somewhat subjective and arbitrary
* The authors need more discussions of (the importance of) Douyin (non-Chinese audiences would have little information)
* While manual checking of the videos minimizes false positives, the authors should consider some form of keyword expansion tool to expand their search query (like https://gking.harvard.edu/files/gking/files/keywordalgorithm.pdf)
* The authors should provide more information about the instructions for manual annotation, classifier performance, etc.

**Questions:**

See the cons section above

**Ethics Review Description:**

-

**Reviewer Confidence:**

4: The reviewer is certain that the evaluation is correct and very familiar with the relevant literature

**Scope:**

3: The work is somewhat relevant to the Web and to the track, and is of narrow interest to a sub-community

---

### Official Review · Reviewer_xDm7 · 2023-12-01

**Novelty:** 7
**Technical Quality:** 6

**Review:**

This paper is fascinating and I enjoyed reading it—what an exciting topic and seemingly strong execution. The writing was easy to understand, and the analysis was straightforward. I appreciated that the author didn't unnecessarily increase the complexity of their analysis when simplicity would tell the story just as well. I also appreciate that the author added limitations and a subsection on ethics. With research like this using publically available data, researchers often neglect this.

I have notes meant to improve readability.

1. Add commas to numbers in tables such as Table 1 "Number of Comments" data to increase legibility.
2. Normalize title casing on subsection headers like subsection 3.1 (which also has a type on the title)
3. Titles for sub-sections 5.5 and 5.6 should be plural.

**Questions:**

Could you make the subset of the data you used available on some dataverse?

**Ethics Review Description:**

I selected No.

**Reviewer Confidence:**

3: The reviewer is confident but not certain that the evaluation is correct

**Scope:**

4: The work is relevant to the Web and to the track, and is of broad interest to the community

---

### Decision · Program_Chairs · 2024-01-22

**Decision:**

Accept (Oral)

**Comment:**

Quoting fAja's adequate paper summary: "The paper examines discussions of four interracial relationship types (BM-CW, BW-CM, WM-CW, WW-CM) focused on Douyin. It capitalizes on access to IP geolocation data to shed light on regional differences in users' attitudes and relevant their correlates such as GDP, population, and openness to Western cultures at an aggregate level."

 Overall, there was strong interest, if not excitement, concerning the topic studied as, usually, xenophobia is studied as a purely Western phenomenon. Also on the technical side, the methods chosen seem adequate and reviewers largely viewed the analysis as solid. In particular given that Reviewer fAja, who had the most specific technical concerns, acknowledged many of the author comments as helpful, but then seemed to fail to update their review and scores.

 It would be good to slightly extend the discussion of ethical concerns. At the moment, the authors mention that the data used (of which a sample will be shared) is anonymized. But there is ample literature showing that removing user IDs does not necessarily ensure anonymity. E.g., depending on the platform, it could be possible to search for the text in a comment, or to use other contextual cluse to de-anonymize a subset of users. This might or might not be problematic, and it does not necessarily preclude the publication of the paper. But discussing how/if such de-anonymization might be possible, and what the implications might be, would strengthen the paper and make it further comply with community norms. The final version should also include details such as the IRB number.

 For a future/camera-ready version it would be good to explicitly mention that the exclusive scope is heterosexual couples.